# A Multi-In-Single-Out Network for Video Frame Interpolation without optical flow

## Abstract

In general, deep learning-based video frame interpolation (VFI) methods have predominantly focused on estimating motion vectors between two input frames and warping them to the target time. While this approach has shown impressive performance for linear motion between two input frames, it exhibits limitations when dealing with occlusions and nonlinear movements. Recently, generative models have been applied to VFI to address these issues. However, as VFI is not a task focused on generating plausible images, but rather on predicting accurate intermediate frames between two given frames, performance limitations still persist. In this paper, we propose a multi-in-single-out (MISO) based VFI method that does not rely on motion vector estimation, allowing it to effectively model occlusions and nonlinear motion. Additionally, we introduce a novel motion perceptual loss that enables MISO-VFI to better capture the spatio-temporal correlations within the video frames. Our MISO-VFI method achieves state-of-the-art results on VFI benchmarks Vimeo90K, Middlebury, and UCF101, with a significant performance gap compared to existing approaches.

## 1 Introduction

Video frame interpolation (VFI) is the task of predicting intermediate frames between given input frames. In general, VFI methods (Jiang et al., 2018; Kong et al., 2022; Bao et al., 2019; Shen et al., 2020; Reda et al., 2022; Shangguan et al., 2022) begin by estimating the motion vectors between two input frames, and then they generate a target intermediate frame using this information. However, motion vector-based VFI methods struggle with nonlinear modeling, necessitating additional refinement (Argaw & Kweon, 2022) or multi-frame modeling (Hu et al., 2022) after motion vector warping for VFI. Despite these efforts, VFI tasks involving occlusions and nonlinear motion still face limitations due to their reliance on motion vectors.

Recently, many score-based generative models (Voleti et al., 2022; Danier et al., 2023) have been applied to VFI; however, their primary objective is to generate a diverse range of plausible intermediate frames, rather than focusing on the prediction of accurate intermediate frames. Consequently, their focus on diversity is not beneficial for VFI tasks, which necessitate the generation of accurate intermediate frames. Thus, the ideal VFI model should be capable of effectively modeling both linear and nonlinear motion while generating accurate intermediate frames.

In this paper, we propose an implicit Multi-In-Single-Out (MISO) based VFI model designed to generate accurate intermediate frames that are robust against nonlinear motion. MISO-VFI predicts a target intermediate frame by taking both a prior frame clip and a subsequent frame clip as input, enabling it to capture nonlinear motion effectively. Additionally, unlike existing explicit video frame interpolation methods, our approach employs an implicit structure for the target time point $t$, enabling the prediction of continuous intermediate frames at various $t$. Finally, as the VFI task necessitates a model capable of capturing the surrounding context or video motion, we propose a novel motion perceptual loss that aids in achieving this objective. Our proposed MISO-VFI achieves state-of-the-art performance on all common VFI benchmark datasets.

Our work makes the following contributions:

- We propose a VFI model with a multi-in-single-out structure capable of predicting nonlinear motion.

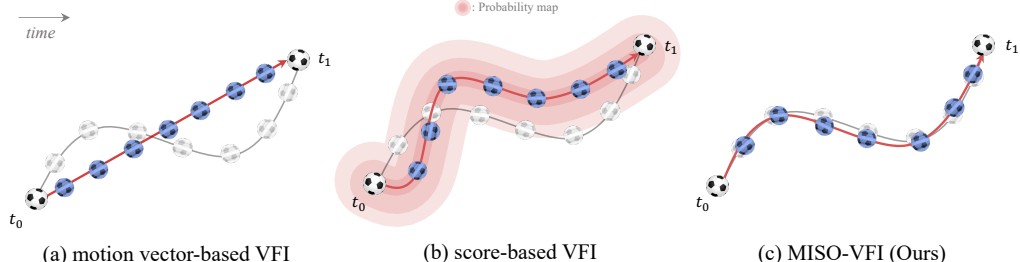

Figure 1: Comparing key VFI development directions: The white balls are the example of the input of the VFI, the shadow region represents the movement of the ball between $t_0$ and $t_1$, and the blue balls represent an example of inference of each method. (a) motion vector-based VFI, widely used but inaccurate for nonlinear motion and occlusion; (b) score model-based VFI, generates diverse frames but struggles with accuracy; (c) our MISO-VFI, generates a single accurate intermediate frame without motion vectors.

- We introduce the video motion perceptual loss, a perceptual loss specifically designed for the video domain.
- We validated our method on various VFI benchmark datasets and achieved state-of-the-art performance.

## 2 RELATED WORK

### 2.1 MOTION VECTOR BASED VFI

Given two consecutive frames $I_1$ and $I_2$, the goal is to generate an intermediate frame $I_t$ at time $t \in (0, 1)$ by estimating the optical flow $F_{1 \to 2}$ and $F_{2 \to 1}$, and using them to warp the input frames. Let $I_t^1$ and $I_t^2$ be the intermediate frames warped from $I_1$ and $I_2$, respectively. The optical flow estimation can be represented as a function $f$:

$$F_{1 \to 2} = f(I_1, I_2), \quad F_{2 \to 1} = f(I_2, I_1). \tag{1}$$

A warping function $w$ is used to warp the input frames towards the intermediate frame:

$$I_t^1 = w(I_1, F_{1 \to 2}, t), \quad I_t^2 = w(I_2, F_{2 \to 1}, 1 - t). \tag{2}$$

The final interpolated frame $I_t$ can be obtained by blending $I_t^1$ and $I_t^2$ using a blending function $b$:

$$\hat{I}_t = b(I_t^1, I_t^2, \alpha), \tag{3}$$

where $\alpha$ is a blending weight that controls the contribution of each warped frame to the final result. In deep learning-based methods, the functions $f$, $w$, and $b$ are implemented as differentiable neural network layers, allowing the entire pipeline to be trained end-to-end. The optical flow estimation function $f$ is typically a deep convolutional neural network (CNN), while the warping function $w$ and blending function $b$ can be implemented using differentiable warping and blending layers, respectively.

For instance, in Super SloMo (Jiang et al., 2018), the optical flow estimation network is designed based on the U-Net(Ronneberger et al., 2015) architecture, while the warping and blending are performed using differentiable bilinear sampling and a sub-network that computes the blending weights, respectively. In SepConv (Niklaus et al., 2017), the warping function is replaced with a learnable, spatially-adaptive convolution, which computes a set of filters for each pixel to interpolate high-quality motion trajectories.

However, since these methods fundamentally rely on the optical flow estimation function $f$, they may struggle to generate accurate intermediate frames when the temporal gap between $I_1$ and $I_2$ is large or when certain content is not present in both $I_1$ and $I_2$. Therefore, the video frame interpolation task becomes increasingly similar to the video frame prediction task (Liang et al., 2017; Liu et al., 2018; Reda et al., 2018) as the temporal interval between input frames grows.

## 2.2 GENERATE MODEL BASED VFI

In recent years, various generative models $G(.)$ have been proposed for video frame interpolation, capturing the distribution of video frames $p_\theta(I)$. These models can be parameterized by $\theta$ and often involve learning a latent variable or a denoising process, depending on the specific methods used.

$$\hat{I}_t = G(I_1, I_2; \theta, z), \tag{4}$$

**VAE-based models**: Variational Autoencoders (VAEs) (Kingma & Welling, 2013) have been applied to video frame interpolation tasks, such as VideoGPT (Yan et al., 2021). VAE-based models learn a latent variable $z$ and optimize the reconstruction error between generated and ground truth intermediate frames while constraining the latent space to have a smooth distribution.

**GAN-based models**: Generative Adversarial Networks (GANs) (Goodfellow et al., 2020) have also been employed for video frame interpolation, as in DIGAN (Yu et al., 2022). GAN-based models consist of a generator and a discriminator, which compete to generate realistic intermediate frames and distinguish between real and generated frames, respectively. In this case, the latent variable $z$ is used to generate diverse intermediate frames that can deceive the discriminator.

**Score-based models**: Recently, score-based methods have been proposed for video frame interpolation, such as the diffusion model in (Voleti et al., 2022; Ho et al., 2020; Song et al., 2020). These models iteratively corrupt a ground truth intermediate frame with noise through a forward diffusion process and then recover the intermediate frame through a reverse diffusion process that progressively denoises the corrupted frame. A deep neural network is employed to estimate the noise at each diffusion step, taking into account temporal information from input frames.

Despite these efforts, generative model-based video frame interpolation methods do not predict accurate intermediate frames due to the randomness introduced by the random value $z$ or Gaussian noise $z$. Instead, they generate a variety of plausible frames. The notion of "plausible" frames implies that the generated images may not be accurate representations close to the Ground Truth (GT). In other words, this means they might incur high loss in pixel-wise metrics such as Mean Squared Error (MSE) or Mean Absolute Error (MAE).

In this paper, we observe that as the time interval between $I_1$ and $I_2$ increases, video frame interpolation becomes more challenging and shares similarities with the video frame prediction (Seo et al., 2023b) task in terms of objectives and characteristics. Motivated by the need to obtain accurate intermediate frames without relying on optical flow, we propose a *Solid Multi-In-Single-Out Network* for video frame interpolation.

## 2.3 MULTIPLE FRAME BASED VFI

Various studies have been proposed that use multiple frames to capture non-linear motion, similar to our MISO VFI (Jiang et al., 2018; Liu et al., 2020). While these studies demonstrated good performance in capturing non-linear motion, all of them relied on generating optical flow and using a warping approach, which showed weak performance in cases of disappearing or obscured objects

Recently, in alignment with the objectives addressed by MISO VFI, *FLAVR* (Kalluri et al., 2023) was introduced. This approach utilizes multiple frames without the dependence on optical flow. Nevertheless, its architecture is predicated on ingesting multiple frames and subsequently producing multiple frames as output. Contrary to conventional Video Frame Interpolation (VFI) methodologies, it is constrained to generating a predetermined fixed frame solely between $I_1$ and $I_2$.

In contrast, our model, MISO-VFI, despite accepting multiple frames as input, operates as an implicit model with respect to $t$. Consequently, analogous to prior flow-based VFI techniques, it possesses the capability to interpolate frames indiscriminately between $I_1$ and $I_2$.

## 3 METHOD

In this section, we first describe the detailed structure of MISO-VFI. Next, we provide an in-depth explanation of the motion perceptual loss, which endows MISO-VFI with the ability to capture

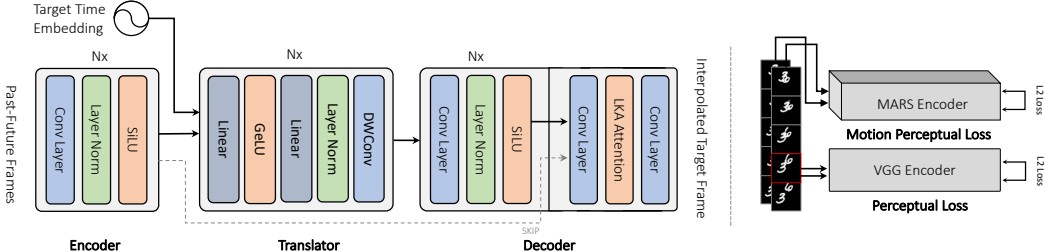

Figure 2: MISO-VFI's T-shaped (Seo et al., 2023b) architecture consists of an encoder, predictor, and decoder. The target time $t$ is incorporated via sinusoidal positional embedding and an MLP layer. The model employs 2D perceptual loss, motion perceptual loss, and MSE loss.

motion within a video clip. Finally, we present a comprehensive summary of the input and output objective functions for MISO-VFI.

### 3.1 MULTI-IN-SINGLE-OUT FOR VIDEO FRAME INTERPOLATION

MISO-VFI is an implicit model that generates various time-point interpolation targets $t$ within a single model. Therefore, MISO-VFI generates an interpolated image $\hat{I}_t$ corresponding to ground truth $I_t$ by receiving past frames $I_p = I_{p_1}, I_{p_2}, \ldots, I_{p_n}$, future frames $I_f = I_{f_1}, I_{f_2}, \ldots, I_{f_n}$, and the interpolation target $t$. Given a MISO-VFI model $F_\theta(\cdot)$, where $\theta$ represents the model's parameters, the objective is to train $\theta$ for the following purpose:

$$\theta^* = \arg\min_\theta \mathcal{L}(F_\theta(I_p, I_f, t), I_t),\tag{5}$$

here, $\mathcal{L}(\cdot)$ denotes the loss function that measures the discrepancy between the model's predictions and the ground truth. The goal is to minimize this loss by optimizing the model parameters $\theta$.

As illustrated in Eq 5, MISO-VFI does not rely on optical flow, unlike traditional optical flow-based VFI methods. Consequently, it avoids the constraints imposed by the optical flow estimation function, which heavily depends on brightness constancy and spatial smoothness. Moreover, unlike score-based models, GANs, and VAE-based methods, the objective function of MISO-VFI is a direct method that does not rely on intermediate representations or latent variables. This property can potentially lead to improved performance in video frame interpolation tasks.

### 3.2 TARGET TIME EMBEDDING

To generate a frame of specific time $t$, we employ the target time embedding module to MISO-VFI. According to Sitzmann et al. (2020), periodic functions are effective for embedding continuous coordinate values. We use the SIREN (Sitzmann et al., 2020) activation function for periodic embeddings about target time index $t$.

### 3.3 PERCEPTUAL LOSS

**Preliminaries of perceptual loss** Perceptual loss is a powerful approach to capturing high-level semantic features in images and videos, which can lead to better visual quality and improved performance in various computer vision tasks.

Given an interpolated image $\hat{I}$ by MISO-VFI, a pre-trained feature extraction model $\Phi$, and a target image $I$, the perceptual loss $\mathcal{L}_p$ can be defined as the distance between the high-level features extracted from the two images:

$$\mathcal{L}_p(\hat{I}, I) = \sum_{i=1}^{n} w_i \cdot ||\Phi_i(\hat{I}) - \Phi_i(I)||_2^2,\tag{6}$$

Table 1: Quantitative comparison of our method with baseline methods on the UCF101, Vimeo90K, and Middlebury datasets. The table presents performance metrics (PSNR, and SSIM) for each method, highlighting the performance of our approach. For each experiment, Vimeo90K and Middlebury used 2-3-2 (P-I-F) setting and UCF101 used 5-5-5 setting.

| Method | Vimeo90K (2-3-2) | | Middlebury (2-3-2) | | UCF101 (5-5-5) | | |
|---|---|---|---|---|---|---|---|
| | PSNR | SSIM | PSNR | SSIM | PSNR | SSIM | Params |
| BMBC (Park et al., 2020) | 27.890 | 0.888 | 24.189 | 0.786 | 26.400 | 0.862 | 11.004M |
| XVFI (Sim et al., 2021) | 25.812 | 0.819 | 20.380 | 0.640 | 28.307 | 0.870 | 5.626M |
| ABME (Park et al., 2021) | 26.399 | 0.829 | 21.825 | 0.687 | 25.903 | 0.831 | 18.543M |
| IFRNet (Kong et al., 2022) | 27.550 | 0.881 | 19.286 | 0.834 | 30.972 | 0.849 | 4.959M |
| EMA-VFI (Zhang et al., 2023) | 27.011 | 0.897 | 23.017 | 0.813 | 28.748 | **0.929** | 65.662M |
| MCVD (Voleti et al., 2022) | 23.935 | 0.786 | 20.539 | 0.760 | 22.138 | 0.788 | 739.4M |
| MISO-VFI (Ours) | **37.196** | **0.955** | **38.72** | **0.979** | **36.239** | 0.928 | 20.289M |

where $\Phi_i(\hat{I})$ denotes the feature maps extracted from the $i^{th}$ layer of the feature extraction model $\Phi$, $n$ is the number of layers used for feature extraction, and $w_i$ represents the weight of the $i^{th}$ layer in the overall perceptual loss.

Although perceptual loss has been extensively applied in the video domain, it primarily focuses on capturing high-level semantic features in individual frames, which may not effectively capture the motion information present in videos. As a result, this approach can be considered suboptimal for tasks where motion information plays a critical role, such as video frame interpolation.

**Motion perceptual loss** Inspired by image perceptual loss, we propose a motion perceptual loss that is specifically designed to capture motion in video tasks, providing a more effective approach for these scenarios. To achieve this, the motion perceptual loss adopts a 3D-ResNext architecture that leverages 3D convolution to better encode spatio-temporal information. In contrast to image perceptual loss, which typically uses ImageNet-1k(Russakovsky et al., 2015) pre-trained weights, our motion perceptual loss employs a model pre-trained on the Kinetics-400(Kay et al., 2017) action recognition dataset, allowing it to effectively capture motion-related features.

Furthermore, to place a stronger emphasis on motion, we utilize Optical Flow Hallucination 3D-ResNext in our motion perceptual loss. Optical Flow Hallucination 3D-ResNext is a technique that enables the model to focus on motion even when only RGB inputs are provided. This is achieved by first distilling the features of an action recognition model, which has been trained with optical flow information, when training the RGB model. By incorporating this approach, our motion perceptual loss is better equipped to capture and represent the motion information inherent in video tasks. To adapt the perceptual loss for our motion perception loss, we modify the input in Eq 6 to accommodate video clips instead of a single image $\hat{I}$ such as Eq 7.

$$\mathcal{L}_{mp}(I_p, \hat{I}, I, I_f) = \sum_{i=1}^{n} w_i \cdot ||\Phi_i^{3d}(I_p, \hat{I}, I_f) - \Phi_i^{3d}(I_p, I, I_f)||_2^2, \tag{7}$$

### 3.4 SUMMARY OF MISO-VFI

Figure 2 provides a detailed illustration of MISO-VFI. As depicted, MISO-VFI is composed of a T-shaped architecture, consisting of an encoder, a translator, and a decoder. The model employs Mean Squared Error (MSE) loss, image perceptual loss, and motion perceptual loss as its objective functions. Consequently, the final objective function of MISO-VFI, represented by $F(\cdot)$, can be defined as follows:

$$\mathcal{L}(\hat{I}, I) = \alpha \mathcal{L}_{MSE}(\hat{I}, I) + \beta \mathcal{L}_p(\hat{I}, I) + \gamma \mathcal{L}_{mp}(\hat{I}_{total}, I_{total}), \tag{8}$$

where $I$ is the ground truth image, $\alpha$, $\beta$, and $\gamma$ are the weights of the respective loss components. Note that the notation $I_{total}$ represents the combination of $I_{past}$, $I_{target}$, and $I_{future}$ into a single video clip.

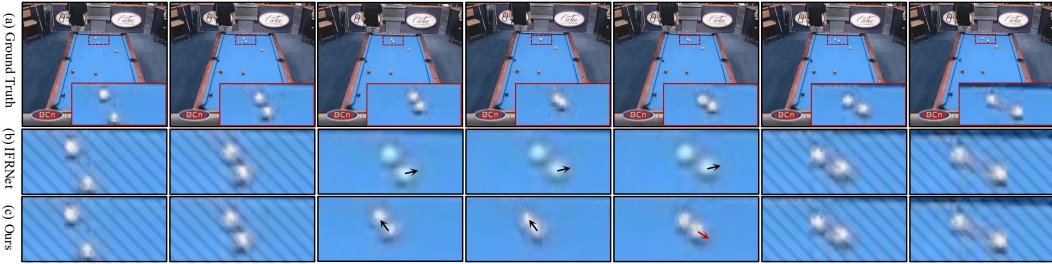

Figure 3: Qualitative comparison of MISO-VFI and previous art, IFRNet, in handling nonlinear motion. Also, the arrow is a vector for the movement of the ball.

## 4 EXPERIMENTS

In this section, we have presented the experimental setup, results, and ablation studies of our proposed MISO-VFI. We evaluated our method on three widely-used benchmark datasets, namely Middlebury (Soomro et al., 2012), Vimeo90K (Xue et al., 2019), and UCF101 (Baker et al., 2011), which cover a broad range of video content and motion patterns. By comparing our method with several state-of-the-art techniques, we demonstrated its superior performance in terms of PSNR and SSIM evaluation metrics. Moreover, the ablation studies offered crucial understanding of the contributions made by individual components in our method, specifically the MISO-VFI architecture and the video perceptual loss. It's noteworthy that the standard video frame interpolation task usually follows a 1-1-1 (**P**ast-**I**ntermediate-**F**uture frames) setup, generating precisely the middle frame. However, in the long-term video frame interpolation framework that we introduce, the quantity of evaluation intermediate frames is expanded. This augmentation serves to better evaluate performance for nonlinear motion and multiple intermediate frames.

### 4.1 EXPERIMENTAL SETUP

**Dataset setup**   Our focus is modeling for non-linearity for long-term video frame interpolation. To achieve this, we find it unreasonable to interpret non-linear motion with only two input images. Consequently, we focus on modeling non-linearity through multiple inputs. For this multi-frame input experiment, we considered a dataset, and fortunately, the septuplet dataset of Vimeo 90K includes seven frames in each video clip, and the UCF101, Middlebury datasets also has two or more frames in each video clip. In addition, in the Vimeo 90K experiment (1-1-1(P-I-F)) in Table 2, we evaluate with the triplet dataset for fair experimental comparison.

**Implementation details**   Our method was implemented using the PyTorch deep learning framework and executed on a system equipped with an NVIDIA A100 40G GPU. We employed the Adam optimizer with $\beta 1 = 0.9$ and $\beta 2 = 0.999$, along with a cosine scheduler without warm-up in all experiments. The learning rate was set to 0.001, and the mini-batch size was 16. To address the learning instability of the model, we applied an exponential moving average (EMA) (Tarvainen & Valpola, 2017), which is not commonly used in existing future frame interpolation models. The EMA model was updated every 10 iterations, with updates starting at the 2,000th iteration. The EMA model update momentum was set to 0.995. We conducted experiments on the Vimeo90K, Middlebury, and UCF101 datasets. For Vimeo90K and Middlebury, where each clip had fewer than 8 frames available, we experimented with a Past-Intermediate-Future frame setup of 2-3-2. As for UCF101, since each clip had the full video available, we were able to perform a 5-5-5 experiment.

### 4.2 EXPERIMENTAL RESULTS

**Quantitative results**   Table 1 presents the quantitative experimental results. Our method outperforms baselines, including BMBC (Park et al., 2020), XVFI (Sim et al., 2021), ABME(Park et al., 2021), IFRNet (Kong et al., 2022), EMA-VFI (Zhang et al., 2023), and MCVD (Voleti et al., 2022), across all datasets (UCF101, Vimeo90K, and Middlebury). Notably, our method does not use optical flow for the first time and still achieves high performance compared to other methods. While

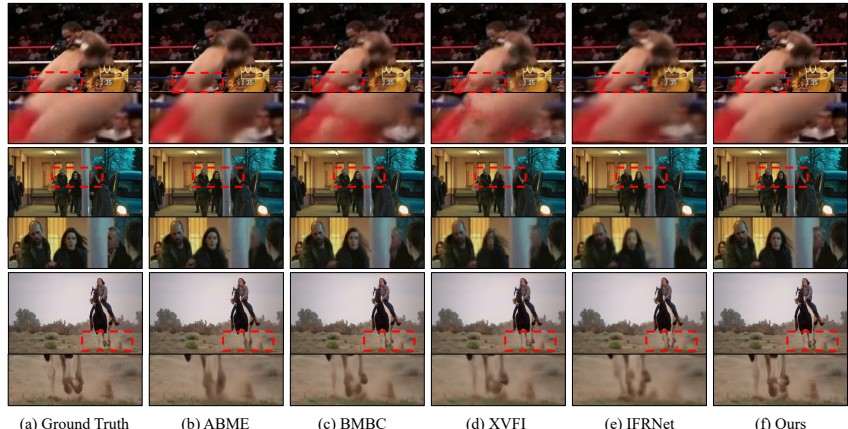

|(a) Ground Truth | (b) ABME | (c) BMBC | (d) XVFI | (e) IFRNet | (f) Ours|

Figure 4: Qualitative comparison results for MISO-VFI in handling occluded.

Table 2: Performing a quantitative performance comparison with MISO-VFI under the conventional VFI experimental setup 1-1-1(P-I-F).

| Method | Vimeo90K-triplet (I=1) | | Middlebury (I=1) | | UCF101 (I=1) | | |
| --- | --- | --- | --- | --- | --- | --- | --- |
| | PSNR | SSIM | PSNR | SSIM | PSNR | SSIM | Params |
| BMBC | 35.01 | 0.976 | 35.22 | 0.941 | 34.32 | 0.920 | 11.004M |
| XVFI | 35.07 | 0.976 | 36.49 | 0.959 | 34.76 | 0.942 | 5.626M |
| ABME | 36.18 | 0.980 | 37.23 | 0.962 | 35.38 | 0.969 | 18.543M |
| IFRNet | 36.20 | 0.980 | 37.50 | 0.963 | 35.42 | 0.969 | 4.959M |
| EMA-VFI | 36.64 | **0.981** | **38.42** | **0.975** | 35.48 | **0.970** | 65.662M |
| MCVD | 25.72 | 0.810 | 20.53 | 0.760 | 22.18 | 0.788 | 739.4M |
| MISO-VFI (Ours) | **38.18** | 0.969 | 36.279 | 0.928 | **38.044** | 0.9591 | 20.289M |

MCVD also does not utilize optical flow, its training purpose is generation-oriented, resulting in lower performance than our prediction-oriented MISO-VFI.

In our empirical analysis, we augmented the number of intermediate frames derived from the original singular frame to three for Vimeo90k, three for Middlebury, and five for UCF101. This was done to underscore the prowess of our model in capturing nonlinear motion. Given this context, one might anticipate that our architecture, meticulously crafted for multi-frame prediction, would manifest exemplary performance. Nevertheless, even when restricted to the generation of a sole intermediate frame—a setting reminiscent of traditional VFI experiments—MISO-VFI surpasses the state-of-the-art across all three datasets.

Specifically, in Tab 2 MISO-VFI achieved scores of 38.504, and 38.044, exceeding EMA-VFI's Vimeo90k score of **36.64**, and EMA-VFI's UCF101 score of **35.48**, respectively. These results indicate that our method not only excels in multi-frame generation but also demonstrates robust performance in generating a single intermediate frame.

Furthermore, MISO-VFI can be applied not only to RGB images but also to various sensor data. Please refer to **Appendix B***(Satellite calibration experiment results)* for this.

**Qualitative results**  We conducted a qualitative comparison of MISO-VFI with other baselines on samples containing occlusions within the Vimeo and UCF101 datasets to assess the robustness of MISO-VFI in handling nonlinear motion and occlusions.

Figure 3 illustrates a qualitative comparison between MISO-VFI and previous art concerning nonlinear motion scenarios. We fed into our model past and future frames featuring a billiard ball exhibiting nonlinear motion and tasked it with generating intermediate frames. As the figure demonstrates, MISO-VFI accurately reproduces the trajectory of the billiard balls colliding and diverging, while the optical flow-based previous methods, assuming linear motion, inaccurately represent the direction of movement. These experimental outcomes suggest a promising potential for the implementation of our methodology in contexts such as sports videos, where accurate modeling of nonlinear motion is paramount.

Table 3: Experimental results comparing performance changes according to the number of input frames and output frames in the SMMNIST dataset.

| Method | Condition | | | Metric | |
|---|---|---|---|---|---|
| | P | I | F | PSNR | SSIM |
| MCVD | 5 | 5 | 5 | 27.693 | 0.941 |
| MCVD | 5 | 10 | 5 | 25.324 | 0.909 |
| MCVD | 5 | 15 | 5 | 21.986 | 0.860 |
| MISO-VFI | 5 | 5 | 5 | **42.286** | **0.995** |
| MISO-VFI | 5 | 10 | 5 | 41.129 | 0.973 |
| MISO-VFI | 5 | 15 | 5 | 40.344 | 0.953 |

(a) Comparison of VFI performance based on the number of output frames

| Method | Condition | | | Metric | |
|---|---|---|---|---|---|
| | P | I | F | PSNR | SSIM |
| MCVD | 1 | 5 | 1 | 14.293 | 0.741 |
| MCVD | 3 | 5 | 3 | 23.985 | 0.884 |
| MCVD | 7 | 5 | 7 | 28.109 | 0.943 |
| MISO-VFI | 1 | 5 | 1 | 39.293 | 0.949 |
| MISO-VFI | 3 | 5 | 3 | 41.202 | 0.990 |
| MISO-VFI | 7 | 5 | 7 | **42.870** | **0.991** |

(b) Comparison of VFI performance based on the number of input frames

Figure 6 presents the qualitative analysis results of MISO-VFI for occluded or newly created objects. As demonstrated in figure 6 unlike other baselines, MISO-VFI properly predicts objects that appear after being occluded. These experimental results suggest that MISO-VFI possesses the capability to handle occlusion and perform nonlinear modeling effectively.

## 4.3 ABLATION STUDIES

In this section, we perform a thorough analysis of the performance of MISO-VFI on the Stochastic Moving MNIST(SMMNIST) dataset (Srivastava et al., 2015; Denton & Fergus, 2018), examining the impact of varying the number of inputs and outputs. Additionally, we conduct an ablation study on 3D-perceptual loss. Note that in this section, $P$ stands for past, $I$ stands for intermediate, and $F$ stands for future.

**Input&Output condition dependency** MISO-VFI is an implicit model that takes multiple frames as input and generates an intermediate frame at the target time $t$. As the length of the target $T$ increases within the same training epoch, the opportunity to train each time point $t$ decreases. Table 3-(a) presents a performance comparison according to the length of $T$ within the same epoch (2,000). As the table shows, performance decreases as the length of $T$ increases. However, the performance degradation in our method is relatively small compared to MCVD. This is because MCVD employs auto-regressive inference, causing errors to accumulate and performance to drop as $T$ lengthens.

On the other hand, our MISO-VFI employs a non-auto regressive approach that does not accumulate errors. The observed decrease in performance is primarily due to the reduced opportunity to train each time point $t$ when $T$ increases within the same epoch. (For more detailed experimental results, please refer to the supplementary material.)

The results in Tab 3-(b) illustrate the performance changes according to the number of input frames. As the table indicates, performance decreases as the number of input frames is reduced. Conversely, as the number of inputs increases, the performance improves, but at the cost of a significant increase in computational complexity. These experimental results align with existing VFI approaches that utilize multiple frames.

**Impact of 3D-perceptual loss** We conducted a comparative ablation study using 2D-perceptual loss to evaluate the effectiveness of our proposed 3D-perceptual loss. Table 4 presents the results. As shown in the table, both 2D-perceptual loss and 3D-perceptual loss greatly contribute to VFI performance. The highest performance is achieved when 2D-perceptual loss and 3D-perceptual loss are used in conjunction. These results suggest that 3D-perceptual loss has a complementary relationship with 2D-perceptual loss.

Figure 3 presents a performance comparison graph for various combinations of 3D-perceptual loss. As depicted in the figure, our proposed 3D-perceptual loss (hallucinate-TVL1) demonstrates the highest performance. These experimental results suggest that not only the appearance of the 2D image but also the motion of the 3D clip with the added time axis is an important factor in the VFI task.

Table 4: Ablation study results examining the impact of 2D-perceptual loss and motion perceptual loss.

| Component | | Metric | |
|---|---|---|---|
| 2D-Perceptual | 3D-Perceptual | PSNR | SSIM |
| - | - | 40.674 | 0.9943 |
| ✓ | - | 42.121 | 0.9956 |
| - | ✓ | 40.951 | 0.9945 |
| ✓ | ✓ | **42.286** | **0.995** |

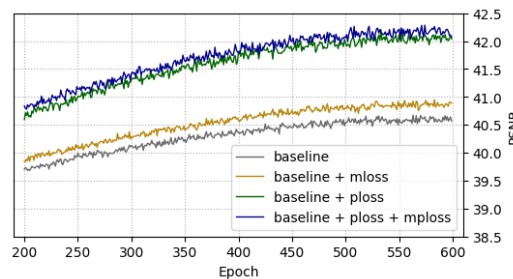

Figure 5: Comparison of convergence rates between 2D-perceptual loss and motion perceptual loss.

## 5 LIMITATION

**Future work**   Given that our MISO-VFI accepts multi-frame input, as shown in  Table 5, its computational demands significantly exceed those of conventional optical flow-based VFI methods. Additionally, our model's implicit time factor, $t$, necessitates substantial training time for convergence. Consequently, future work will focus on devising strategies that are not only computationally efficient, but also consider convergence speed, to enhance the practicality and usability of our model.

**Discussion**   The majority of video frame interpolation tasks are employed for super frame-rate applications, such as upgrading a 25 FPS video to 300 FPS or higher. In these scenarios, generating a plausible image often suffices and exact image reproduction isn't always necessary.  Thus, the concept of producing the "correct" answer through VFI, as proposed by us, might not find broad applicability. Furthermore, our model, when presented with a sequence of multiple frames, is unable to interpolate the first and last frames among the entire frames of the video.  However, through a single-input structure in our model (1-1-1(P-I-F)), interpolation similar to conventional VFI is possible.

Table 5: Flops Comparison in SMMNIST. Our network requires a larger computational load because it inputs multiple frames compared to other networks.

| Method | Condition (P-I-F) | Flops | Shape (T, C, H, W) |
|---|---|---|---|
| IFRNet | (1-1-1) | 0.93G | (2, 3, 64, 64) |
| BMBC | (1-1-1) | 11.1G | (2, 3, 64, 64) |
| ABME | (1-1-1) | 5.7G | (2, 3, 64, 64) |
| XVFI | (1-1-1) | 1.6G | (2, 3, 64, 64) |
| EMA-VFI | (1-1-1) | 0.8G | (2, 3, 64, 64) |
| MCVD | (1-1-1) | 5.56G × 100 steps | (2, 3, 64, 64) |
| MISO-VFI | (1-1-1) | 3.09G | (2, 3, 64, 64) |
| MISO-VFI | (5-1-5) | 22.95G | (10, 3, 64, 64) |

## 6 CONCLUSION

In this paper, we highlighted the limitations of motion vector-based VFI methods, which struggle to model occlusion or nonlinear motion, and proposed a Multi-In-Single-Out (MISO) video frame interpolation structure to address these issues. Additionally, we identified the limitations of 2D-perceptual loss, which only considers 2D information in VFI tasks with a time axis, and introduced a novel 3D-perceptual loss capable of capturing motion. Our proposed MISO-VFI model and 3D-perceptual loss were validated on the Vimeo90K, UCF101, and Middlebury video frame interpolation benchmark datasets, achieving state-of-the-art results across all datasets. Furthermore, we conducted an ablation study of MISO-VFI and 3D-perceptual loss on the Stochastic Moving MNIST dataset to provide deeper insights into VFI. However, our approach still demands significant computation, making real-time applications challenging. We hope that further development of non-optical flow-based structures will occur within the research community.

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

## APPENDIX

## A ADDITIONAL QUALITATIVE RESULTS

Figure 6 provides further qualitative evidence. As demonstrated in the figure, it is evident that MISO-VFI exhibits robustness against small objects and obstructions, which are typically challenging for extracting optical flow. This underscores MISO-VFI's strength in managing complex scenarios, reinforcing its potential for broader application.

## B SATELLITE CALIBRATION EXPERIMENT RESULTS

VFI can be applied not only to RGB but also to various sensors. However, because existing optical flow-based methods were all developed and trained in the RGB space, they are often difficult to apply to other sensors. In such situations, MISO-VFI, which does not use optical flow, can be a good alternative.

To validate MISO-VFI on sensors other than RGB (physical values), we tested it on a geostationary satellite.

In recent times, the resolution of geostationary satellites has seen significant improvements, with 1-2 minute intervals and a pixel size of 1-2 km. Nonetheless, challenges persist in high-resolution analysis for long-term climate applications when endeavoring to incorporate multiple past and present satellites. This is primarily due to the fact that many past geostationary satellites offer lower resolution, typically between 10 to 15 minutes.

To tackle this issue, we have implemented MISO-VFI to bridge the time gap of past satellite intervals with those of the current ones. It's important to highlight that prevalent VFI techniques depend on optical flow and warping, hence they demonstrate weak performance when dealing with emerging or dissipating clouds. Additionally, certain methodologies are incapable of generating optical flow in satellite sensors. The results presented in the Table 6 were obtained by interpolating data between a COMS satellite, which has a temporal interval of 15 minutes, and a GK2A satellite, which operates with a temporal interval of 2 minutes. As the test outcomes suggest, MISO-VFI demonstrated superior performance in IR, WV, and SW sensors, outpacing the existing state-of-the-art techniques. This evidence supports the potential application of MISO-VFI not only in satellite-based scenarios but also in various sensor calibration problems.

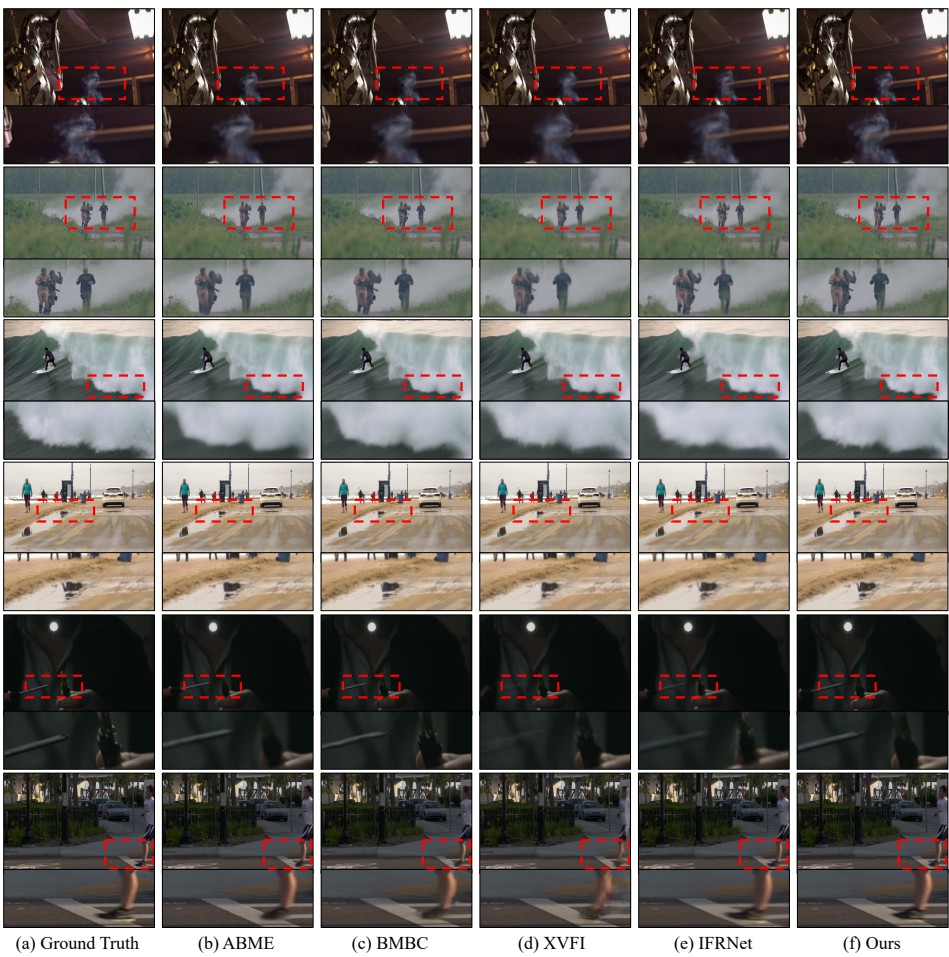

| (a) Ground Truth | (b) ABME | (c) BMBC | (d) XVFI | (e) IFRNet | (f) Ours |

Figure 6: Qualitative comparison results for MISO-VFI.

Table 6: Quantitative experimental results of video frame interpolation of MISO-VFI on the GK2A geostationary satellite weather observation dataset. (IR=infrared, SW=short wave, WV= water vapor)

| Method | Channels | PSNR | SSIM |
|---|---|---|---|
| Linear | IR | 38.667 | 0.745 |
| SSM-T (Vandal & Nemani, 2021) | IR | 43.285 | 0.831 |
| WR-Net (Warp Only) | IR | 44.213 | 0.904 |
| WR-Net (Full) (Seo et al., 2023a) | IR | 46.527 | 0.934 |
| MISO-VFI (Full) | IR | **51.531** | **0.965** |
| Linear | SW | 38.381 | 0.719 |
| SSM-T | SW | 45.935 | 0.827 |
| WR-Net (Warp Only) | SW | 48.237 | 0.922 |
| WR-Net (Full) | SW | 50.538 | 0.936 |
| MISO-VFI (Full) | SW | **52.773** | **0.950** |
| Linear | WV | 43.526 | 0.766 |
| SSM-T | WV | 51.842 | 0.895 |
| WR-Net (Warp Only) | WV | 56.191 | 0.929 |
| WR-Net (Full) | WV | 50.538 | 0.936 |
| MISO-VFI(full) | WV | **55.192** | **0.973** |

# C ARCHITECTURE DETAILS

Table 7 provides a detailed view of the MISO-VFI structure as implemented in the Stochastic Moving MNIST dataset.

Table 7: MISO-VFI architecture details

| Name | Layer | Input Shape | Output Shape |
|------|-------|-------------|--------------|
| **Time Embedding** | | | |
| Name | Layer | Input Shape | Output Shape |
| Input | - | [b, 1] | [b, 1] |
| Time Embedding* | SinusoidalPosEmb | [b, 1] | [b, 64] |
| | Linear | [b, 64] | [b, 256] |
| | GeLU | [b, 256] | [b, 256] |
| | Linear | [b, 256] | [b, 64] |
| | GeLU | [b, 64] | [b, 64] |
| | Linear | [b, 64] | [b, 640] |
| Reshape | Reshape | [b, 640] | [b, 640, 1, 1] |
| **MISO-VFI** | | | |
| Name | Layer | Input Shape | Output Shape |
| Input | - | [b, 10, 1, 64, 64] | [b, 10, 1, 64, 64] |
| Reshape | Reshape | [b, 10, 1, 64, 64] | [b × 10, 1, 64, 64] |
| Encoder** × 4 | Conv2d (kernel=3, stride=[1, 2, 1, 2]) | [b × 10, 1, 64, 64] | - |
| | LayerNorm | - | - |
| | SiLU | - | [b × 10, 64, 16, 16] |
| Reshape | Reshape | [b × 10, 64, 16, 16] | [b, 640, 16, 16] |
| Translator × 3 | Conv2d (kernel=7, padding=3, group=4) | [b, 640, 16, 16] | [b, 640, 16, 16] |
| | Add Time Embeddings* | [b, 640, 16, 16] | [b, 640, 16, 16] |
| | GeLU | [b, 640, 16, 16] | [b, 640, 16, 16] |
| | Conv(kernel=1) | [b, 640, 16, 16] | [b, 640 × 4 , 16, 16] |
| | GeLU | [b, 640 × 4, 16, 16] | [b, 640 × 4, 16, 16] |
| | Conv(kernel=1) | [b, 640 × 4, 16, 16] | [b, 640, 16, 16] |
| Reshape | Reshape | [b, 640, 16, 16] | [b × 10, 64, 16, 16] |
| Decoder-1 × 3 | Pixel Shuffle (kerner=3, stride=1, upscale_factor=[2,1,2]) | [b × 10, 64, 16, 16] | - |
| | LayerNorm | - | - |
| | SiLU | - | [b × 10, 64, 64, 64] |
| Decoder-2 | Concatenate with low-level features** | [b × 10, 64, 64, 64] × 2 | [b × 10, 128, 64, 64] |
| | Pixel Shuffle (kernel=3, stride=1, upscale_factor=2) | [b × 10, 128, 64, 64] | [b × 10, 64, 64, 64] |
| | LayerNorm | [b × 10, 64, 64, 64] | [b × 10, 64, 64, 64] |
| | SiLU | [b × 10, 64, 64, 64] | [b × 10, 64, 64, 64] |
| | Reshape | [b × 10, 64, 64, 64] | [b, 640, 64, 64] |
| | Conv2d (kernel=1) | [b, 640, 64, 64] | [b, 64, 64, 64] |
| | LKA-attention | [b × 64, 64, 64] | [b ×, 64, 64, 64] |
| | Conv2d (kernel=1) | [b, 64, 64, 64] | [b, 1, 64, 64] |
| Output | - | [b, 1, 64, 64] | [b, 1, 64, 64] |

