# OpenReview forum: "A Multi-In-Single-Out Network for Video Frame Interpolation without optical flow"
_ICLR.cc/2024/Conference — Submitted to ICLR 2024_

### Official Review · Reviewer_a9Au · 2023-10-30

**Soundness:** 2 fair
**Presentation:** 2 fair
**Contribution:** 2 fair
**Rating:** 5
**Confidence:** 3

**Summary:**

This paper proposes an implicit Multi-In-Single-Out (MISO) based VFI model designed to generate accurate intermediate frames that are robust against nonlinear motion. It predicts
a target intermediate frame by taking both a prior frame clip and a subsequent frame clip as input, enabling it to capture nonlinear motion effectively. Meanwhile, a video motion perceptual loss is proposed in this paper. The two contributions presented enable the state-of-the-art performance achieved in this paper.

**Strengths:**

1. The proposed structure is simple but effective and capable of achieving state-of-the-art performance.
2. The ablation experiments are more than sufficient to argue for the validity of the two proposed modules.

**Weaknesses:**

1. Will this multiple-input single-output framework have limitations in practical applications? For example, given a T-frame video, how is the first frame or the last stitch of the image processed?
2. EMA-VFI is incorrectly referenced.
3. The length of the METHOD section is too short. The model structure is not presented in the whole paper, and it is not enough to show it only in Figure 2.
4. I think the Table 1 comparison is unfair because the other methods in the table have not been trained on multi-frame video. Therefore Table 1 should not be used as the first table in the experimental chapter. Instead Table 2 will be a fairer comparison.

**Questions:**

Training according to the model structure proposed in this paper requires more video frames. How did this paper modify the dataset so that it meets the training requirements of the structure proposed in this paper?

---

> ### Author Response · Authors · 2023-11-19
>
> First, we express gratitude for the reviewer's review. We focused on modeling non-linear motion using multi-frame input without Optical Flow. As a result, we have designed a model that achieves state-of-the-art performance, as mentioned by the reviewer as a "strength," and it is both simple and effective. We appreciate the opportunity to discuss these results with the reviewer. Below is our rebuttal:
>
> **Q1. Will this multiple-input single-output framework have limitations in practical applications? For example, given a T-frame video, how is the first frame or the last stitch of the image processed?**
>
> Yes, that is a limitation of our method. The first and the last frames cannot be interpolated. We have added this limitation to the text. (However, with a 1-1-1 setting, our method, like typical VFI approaches, can also interpolate the first and last frames.)
>
> **Q2. EMA-VFI is incorrectly referenced.**
>
> Thank you! We have corrected the reference!
>
> **Q3. The length of the METHOD section is too short. The model structure is not presented in the whole paper, and it is not enough to show it only in Figure 2.**
>
> We have thoroughly revised the Methods section to make the methodology more comprehensible and easier to understand. For additional architecture details, please refer to Appendix C.
>
> **Q4. I think the Table 1 comparison is unfair because the other methods in the table have not been trained on multi-frame video. Therefore Table 1 should not be used as the first table in the experimental chapter. Instead Table 2 will be a fairer comparison.**
>
> We appreciate your feedback regarding the comparison in Table 1. You are correct that the other methods listed in the table were not trained on multi-frame video, which may lead to an unfair comparison. Acknowledging this concern, we have decided to reposition Table 2 as the first table in the experimental chapter. Table 2 provides a fairer comparison as it evaluates all methods under the same conditions. We believe this adjustment will present a more balanced and accurate evaluation of the different methods.
>
> **Q5. Training according to the model structure proposed in this paper requires more video frames. How did this paper modify the dataset so that it meets the training requirements of the structure proposed in this paper?**
>
> We did not modify the dataset. Vimeo90K (2-3-2), Middlebury (2-3-2), and UCF101 (5-5-5) officially provide multiple frames. To reduce any misunderstanding, we will detail this information in the implementation details section.

---

> > ### Author Response · Authors · 2023-11-22
> >
> > Dear reviewer,
> >
> > Thank you again for your reviewer's efforts. The reviewer's interest will make our paper more solid. We want to have more discussions with our reviewers, and we still have a lot of time left.  We have responded to the reviewer's questions. If you have any questions or opinions about this, please feel free to fill it out. Thank you.
> >
> > Best regards, Authors of paper 4385

---

### Official Review · Reviewer_CFmx · 2023-10-31

**Soundness:** 3 good
**Presentation:** 3 good
**Contribution:** 2 fair
**Rating:** 5
**Confidence:** 3

**Summary:**

This paper proposes multi-in-single-out MISO structure that targets at frame interpolation task without explicitly estimate optical flow. The target time t is inserted to the network as a time embedding. Based on such structure and a proposed video motion perceptual loss, the proposed model claims advantages on non-linear motions as well as the capability of handling occlusions between frames. Experiments are conducted on the leading benchmarks with ablation studies on the proposed loss function.

**Strengths:**

The proposed contribution is clear, and the paper is easy to follow

**Weaknesses:**

The network structure is straightforwrd, with the loss as the main technique contribution. The loss consists of a MSE loss, image perceptual loss  and motion perceptual loss, with the last one as the claimed contribution. Although these combined loss design seems reasonable, the motion perceptual loss is adapted from the image perceptual loss, which may be considered as delta modifications or improvements.

I'm not directly work on VFI tasks, but I know that there are bunch of work, e.g., burst image processing, that adopts multi-in-sinle-out structure, such as burst image denoising, burst image super-resolution etc. Therefore, claiming multi-in-sinlge-out is not that novel. If it is the frist to the VFI, then it is ok to claim it. This is what I'm not sure. But even so, it is not that novel.

**Questions:**

Optical flow is important to the task of VFI, although it can be omitted, it can also be estimated as an auxiliary inputs upon multi images to the network for assistance. The flow may do something negative, but it also contributes positively on various situations. I have some reservations regarding that flows are completely removed. However, as long as the results are good, I can still buy it.

In table 4, the metric SSIM, with only 2D-perceptual, the SSIM is 0.9956, however, with both 2D and 3D perceptual loss, the SSIM is 0.995, but bolded. Besides, it seems that the 2D-perceptual loss did the most contributions, the contribution of 3D perceptual loss is small.

I do not find any video examples in the supp. which makes me hard to evaluate the real performances.

They saying "as VFI is not a task focused on generating plausible images," is not accurate. The definition of plausible image and accurate image should be explained.

Estimating motions can still deal with the occlusion challenges, based on what type of occlusions, and how large the temporal T is.

More descriptions of how to read the Fig.1, e.g., what are the white and blue balls, what is the shadow regions, should be clearly stated in the figure caption.

---

> ### Author Response · Authors · 2023-11-19
>
> We genuinely appreciate the efforts of the reviewer. As mentioned in the summary, our model interpolates frames for the target time t using multi-frame input without relying on Optical Flow.
>
> Additionally, MISO-VFI is an implicit model, allowing flexible interpolation for t values not explicitly defined during the training process. Additionally, the Motion Perceptual Loss we propose demonstrates additional performance improvement in situations where the performance is already at a high level, and it is remarkably simple. We are pleased to share and discuss these results with the reviewer.
>
> We believe that the reviewer's mentioned "weaknesses" and "questions" will contribute to strengthening our research. Below is our rebuttal:
>
> **Q1. In table 4, the metric SSIM, with only 2D-perceptual, the SSIM is 0.9956, however, with both 2D and 3D perceptual loss, the SSIM is 0.995, but bolded. Besides, it seems that the 2D-perceptual loss did the most contributions, the contribution of 3D perceptual loss is small.**
>
> The reviewer is correct. Indeed, the 2D perceptual loss contributes more significantly than the 3D perceptual loss. However, we believe that an improvement of 0.16 in a context where PSNR is already at 42 is a considerable enhancement. Additionally, since it is compatible with 2D perceptual loss, we see no reason not to use it, even if the performance gain is slight.
>
> **Q2. I do not find any video examples in the supp. which makes me hard to evaluate the real performances.**
>
> We have added video samples through an anonymized link. You can find them here: https://sites.google.com/view/miso-vfi
>
> **Q3. They saying "as VFI is not a task focused on generating plausible images," is not accurate. The definition of plausible image and accurate image should be explained.**
>
> Accurate' means that the image is close to the actual Ground Truth (GT). In other words, it implies a high score in pixel-wise metrics like MSE (Mean Squared Error) or MAE (Mean Absolute Error). On the other hand, 'plausible' refers to high scores in perceptual metrics such as FVD (Fréchet Video Distance) or FID (Fréchet Inception Distance). We have revised the text to reduce any misunderstanding among readers.
>
> **Q4. Estimating motions can still deal with the occlusion challenges, based on what type of occlusions, and how large the temporal T is.**
>
> Estimating motions to address occlusion challenges depends significantly on the type of occlusions and the temporal interval T. For instance, occlusions caused by objects moving across the frame can be more manageable when T is small, as the motion between frames is less complex. However, larger temporal intervals may introduce more significant challenges, especially if the occlusion involves complex movements or transformations. Additionally, the nature of the occluded object, such as its size, shape, and the degree of its overlap with other objects, plays a crucial role in determining the efficacy of motion estimation. Our approach aims to balance these factors to optimize motion estimation under varying occlusion conditions
>
> **Q5. More descriptions of how to read the Fig.1, e.g., what are the white and blue balls, what is the shadow regions, should be clearly stated in the figure caption.**
>
> We have revised Figure 1 in more detail and also added a caption to it.

---

> > ### Author Response · Authors · 2023-11-22
> >
> > Dear reviewer,
> >
> > Thank you again for your reviewer's efforts. The reviewer's interest will make our paper more solid. We want to have more discussions with our reviewers, and we still have a lot of time left.  We have responded to the reviewer's questions. If you have any questions or opinions about this, please feel free to fill it out. Thank you.
> >
> > Best regards, Authors of paper 4385

---

### Official Review · Reviewer_rS3w · 2023-10-31

**Soundness:** 2 fair
**Presentation:** 3 good
**Contribution:** 1 poor
**Rating:** 5
**Confidence:** 5

**Summary:**

This paper aims to tackle the challenge of multi-frame-based video frame interpolation. Based on an existing T-shaped architecture (Seo et al., 2023b), the author introduces a motion perceptual loss that measures the distance between 3D conv features. Experiments on datasets including UCF101, Vimeo90K, and Middlebury datasets are conducted to demonstrate the proposed method's effectiveness.

**Strengths:**

- This paper is simple and easy to read.
- Very good performance are reported by the authors.

**Weaknesses:**

- Hard to find the contributions of this paper. The model architecture is borrowed from an existing work ( though just on arxiv), it is hard to count as this paper's contribution in my opinion.  The only new thing in this work is the motion perceptual loss, which is too trivial and straightforward. Moreover, as shown in Tab.4, the improvement introduced by motion perceptual loss is too minor  (0.16 PSNR).

- Evaluations are not convincing. Tab-1 compares performance for multi-frame-based interpolation. Yet almost all the selected previous SOTA are not optimized for multi-frame input. The used datasets like Vimeo90K, Middlebury, and UCF101 are also very wired for this setting. In general, the reviewer believes the comparisons are unfair and less convincing.

**Questions:**

- What are the training details for experiments in Tab-2?

---

> ### Author Response · Authors · 2023-11-19
>
> Firstly, we express our gratitude for the reviewer's efforts. As noted by the reviewer, our research demonstrates high performance with a straightforward methodology. Additionally, our proposed Motion Perceptual Loss, with its unique originality, has shown additional performance improvements over an already significantly high baseline, even if it may appear to exhibit modest gains.
>
> We are pleased to share this research with the reviewer and believe that the reviewer's efforts will further strengthen our study. Below is our rebuttal:
>
> **Q1. Hard to find the contributions of this paper. The model architecture is borrowed from an existing work ( though just on arxiv), it is hard to count as this paper's contribution in my opinion. The only new thing in this work is the motion perceptual loss, which is too trivial and straightforward. Moreover, as shown in Tab.4, the improvement introduced by motion perceptual loss is too minor (0.16 PSNR).**
>
> To the best of our knowledge, MISO-VFI is the first work to achieve state-of-the-art in the VFI field without using optical flow. Furthermore, the architecture that embeds lead time $t$ in VFI is also a novel aspect of our work. Lastly, an improvement of 0.16 in PSNR, especially in a context where PSNR values are already between 38 and 39, represents a significant enhancement in performance
>
> **Q2. Evaluations are not convincing. Tab-1 compares performance for multi-frame-based interpolation. Yet almost all the selected previous SOTA are not optimized for multi-frame input. The used datasets like Vimeo90K, Middlebury, and UCF101 are also very wired for this setting. In general, the reviewer believes the comparisons are unfair and less convincing.**
>
> Table 1 does not represent an entirely fair comparison. To address this issue, we conducted experiments under a fair 1-1-1 setup, as shown in Table 2, and evaluated the performance accordingly. As indicated in Table 2, our approach still achieves state-of-the-art results. Moreover, Table 1 was designed to demonstrate the significant advantage of using multi-frame input in VFI
>
> **Q3. What are the training details for experiments in Tab-2?**
>
> We have added detailed experimental information for the experiments in Tables 1 and 2 in the ‘Dataset setup’ section of the text.

---

> > ### Author Response · Authors · 2023-11-22
> >
> > Dear reviewer,
> >
> > Thank you again for your reviewer's efforts. The reviewer's interest will make our paper more solid. We want to have more discussions with our reviewers, and we still have a lot of time left.  We have responded to the reviewer's questions. If you have any questions or opinions about this, please feel free to fill it out. Thank you.
> >
> > Best regards, Authors of paper 4385

---

### Official Review · Reviewer_WQaG · 2023-11-01

**Soundness:** 2 fair
**Presentation:** 2 fair
**Contribution:** 1 poor
**Rating:** 5
**Confidence:** 5

**Summary:**

This paper introduces a novel multi-frame video frame interpolation approach that incorporates explicit non-linear motion estimation. Additionally, the authors propose the use of a perceptual loss to enhance the visual quality of the interpolated frames.


I appreciate your response. Upon reviewing the supplementary results, I am pleased with their quality, particularly in challenging scenarios. As a result, I have raised my final score. However, I still find it challenging to accept this manuscript. I suggest that the authors revise their paper by offering more comprehensive explanations of their motivation and conducting additional analytical experiments.

**Strengths:**

1. The experimental results provide evidence of the superior performance of the proposed model when compared to existing methods.
2. The utilization of a different pre-trained model for motion perceptual loss enhances the overall effectiveness of the approach.

**Weaknesses:**

1. The approach of using an increased number of input frames to interpolate a single intermediate image may lack novelty in its overall concept.
2. The straightforward substitution of the pre-trained model for perceptual loss can be seen as a technical workaround, and there is a lack of clarity regarding the process of constraining a single interpolated image to achieve 3D-perceptual loss.
3. There is a lack of visual analysis provided for the non-linear motion estimation.
4. The quantitative comparison conducted on the Vimeo90K septuplet dataset is not commonly used. It is recommended to provide the results on the triplet benchmark using the 1-1-1 setup for a more standard comparison.

**Questions:**

see the weaknesses

---

> ### Author Response · Authors · 2023-11-19
>
> First of all, we genuinely appreciate the reviewer’s efforts spent on our work. Our model demonstrates superior performance compared to existing models. Additionally, we have successfully introduced Perceptual Loss in the video domain and video frame interpolation. We are pleased to share these advantages with reviewers.
>
> We have thoroughly reviewed and considered your review. The reviewer's feedback will contribute to strengthening our study. Below is our rebuttal:
>
> **Q1. The approach of using an increased number of input frames to interpolate a single intermediate image may lack novelty in its overall concept.**
>
> While structures that take multiple frames as input and output a single frame already exist, to our knowledge, this is the first paper to achieve state-of-the-art in Video Frame Interpolation (VFI) without using optical flow.
>
> **Q2. The straightforward substitution of the pre-trained model for perceptual loss can be seen as a technical workaround, and there is a lack of clarity regarding the process of constraining a single interpolated image to achieve 3D-perceptual loss.**
>
> To operationalize 3D perceptual loss, we concatenate \( I_{p} \), \( \hat{I} \), and \( I_{f} \) as inputs to \( \Phi \) for loss calculation. We have revised the text to clarify the implementation of 3D perceptual loss.
>
> **Q3. There is a lack of visual analysis provided for the non-linear motion estimation.**
>
> Additional video results are provided through an anonymized link for a more comprehensive visual analysis of non-linear motion estimation. https://sites.google.com/view/miso-vfi
>
> **Q4. The quantitative comparison conducted on the Vimeo90K septuplet dataset is not commonly used. It is recommended to provide the results on the triplet benchmark using the 1-1-1 setup for a more standard comparison.**
>
> We have altered our experiments to use the 1-1-1 setup as a triplet benchmark. Our model still achieves state-of-the-art results in this more standard comparison.

---

> > ### Author Response · Authors · 2023-11-22
> >
> > Dear reviewer,
> >
> > Thank you again for your reviewer's efforts. The reviewer's interest will make our paper more solid. We want to have more discussions with our reviewers, and we still have a lot of time left.  We have responded to the reviewer's questions. If you have any questions or opinions about this, please feel free to fill it out. Thank you.
> >
> > Best regards, Authors of paper 4385

---

### Author Response · Authors · 2023-11-23

We deeply appreciate the reviewer's efforts in the academic field. Thanks to the reviewer, the quality of our paper will greatly improve.

We have recognized that conventional Video Frame Interpolation models fail to interpret nonlinear motions, so we propose A Multi-In-Single-Out VFI (MISO-VFI) that does not use optical flow for this matter. To the best of our knowledge, MISO-VFI is the first work to achieve SOTA in the VFI field without using optical flow. Also, our work is flexibly adaptable to standard VFI (PIF 1-1-1) as well as in multi-in-single-out manner. Furthermore, even though we already established a strong high-scored baseline, we proposed additional Motion Perceptual Loss which contributed to further performance improvement.

Table 1 shows that the MISO-VFI produced better results over other methods in a Multi-In-Single-Out manner. Moreover, we also demonstrated that the MISO-VFI got the upper hand over other methods in the standard VFI setting, as shown in Table 2. Finally, we show the effect of interpreting nonlinearity in Figures 1, 4, and 6 (in Appendix) and our anonymized link https://sites.google.com/view/miso-vfi.

Best regards, Authors of paper 4385

---

### Meta-Review · Area_Chair_C7v2 · 2023-12-07

**Metareview:**

This paper presents a method for video frame interpolation which does not require the estimation of optical flow. The main motivation to do so, is that optical flow is limiting with there is occlusion. The approach achieves good empericaly performance and outperforming prior works. However, the approach isn't particularly well motivated and certain statements are confusing, as pointed out by the reviewers. For example, the abstract states "VFI is not a task focused on generating plausible images, but rather on predicting accurate intermediate frames between two given frames, performance limitations still persist." Why are "plausible images" not accurate? Finally, several reviewers mention that the comparision is not entiring fair due to the change in setting. While the authors have responded with Table 2, the paper can be strengthened by following and comparing to **all** reported settings in existing works.

**Justification For Why Not Higher Score:**

A more comprehensive empirical comparison following prior works' setting is necessary to properly evaluate the proposed method. The AC also recommends the authors to polish the writing and better explain some of the approach details.

**Justification For Why Not Lower Score:**

N/A

---

### Decision · Program_Chairs · 2024-01-16

Reject